# Investigation of the Microstructure of Sintered Ti–Al–C Composite Powder Materials under High-Voltage Electrical Discharge

**DOI:** 10.3390/ma16175894

**Published:** 2023-08-29

**Authors:** Rasa Kandrotaitė Janutienė, Darius Mažeika, Jaromír Dlouhý, Olha Syzonenko, Andrii Torpakov, Evgenii Lipian, Arūnas Baltušnikas

**Affiliations:** 1Department of Production Engineering, Kaunas University of Technology, 44249 Kaunas, Lithuania; darius.mazeika@ktu.lt; 2COMTES FHT a.s., 33441 Dobřany, Czech Republic; jaromir.dlouhy@comtesfht.cz; 3Institute of Pulse Processes and Technologies, National Academy of Science of Ukraine, 01030 Kyiv, Ukraine; sizonenko43@rambler.ru (O.S.); torpakov@gmail.com (A.T.); olgasizonenko43@gmail.com (E.L.); 4Lithuanian Energy Institute, 44403 Kaunas, Lithuania; arunas.baltusnikas@lei.lt

**Keywords:** electron microscopy, composites, powder methods, spark plasma sintering, high-voltage electrical discharge

## Abstract

Dispersion-hardened materials based on TiC–AlnCn are alloys with high heat resistance, strength, and durability that can be used in aircraft and rocket technology as a hard lubricant. The titanium-rich composites of the Ti–Al–C system were synthesized via the spark plasma sintering process. Composite powder with 85% of Ti, 15% of Al, and MAX-phases was processed using high-voltage electrical discharge in kerosene at a specific energy of 25 MJ kg^−1^ to obtain nanosized particles. This method allows us to analyze the most efficient, energy saving, and less waste-generating technological processes producing materials with improved mechanical and physical properties. An Innova test indentation machine was used to determine the hardness of the synthesized composites. The microhardness of Ti–Al–C system samples was determined as approximately 500–600 HV. Scanning electron microscopy and energy-dispersive X-ray spectroscopy were performed to identify the hard titanium matrix reinforced by intermetallic phases and the clusters of carbides. Three types of reinforcing phases were detected existing in the composites—TiC, Al_4_C_3_, and Al_3_Ti, as well as a matrix consisting of α- and β-titanium. The lattice parameters of all phases detected in the composites were calculated using Rietveld analysis. It was determined that by increasing the temperature of sintering, the amount of aluminum and carbon increases in the carbides and intermetallic phases, while the amount of titanium decreases.

## 1. Introduction

Metal matrix composites (MMCs) combine the properties of their components, namely the high plasticity of their metal matrix with the rigidity and hardness of non-metallic strengthening particles, which leads to excellent mechanical performance. Titanium matrix composites (TMCs) are one of the prospective MMCs types, as they have outstanding properties, including relatively low density, as well as high levels of specific strength and corrosion resistance. Currently, TMVs are widely used in different industrial fields, including biomedicine and the automotive and aerospace industries [1,2,3].

Dispersion-hardened composites based on titanium–aluminum–carbon (Ti–Al–C) alloy systems are used in the aviation and aerospace industries owing to their combined properties of metals, which are ductile and fracture resistant, and ceramics, resulting in strong materials with high elastic modulus. They consist of a titanium matrix and hardening inclusions, such as titanium carbides, titanium–aluminum–carbon compounds, and fullerenes (C_60_ and C_70_), which decrease the coefficient of friction. In addition to their high strength, such nanocomposites present high thermal resistance over a wide range of temperatures and a low coefficient of friction. Hence, these composites are considered advanced materials.

The properties of the products are highly dependent on the quality of the blend. The failure to follow the optimum mixture preparation mode leads to the appearance of individual large grains or their splices in the structure of the sintered materials, which are then pockets of brittle fracture. Moreover, large metal inclusions lead to the higher porosity of the material. Co-milling-mixing of the titanium carbide powder and metal component in ball mills is the main method of preparation of initial mixtures to obtain composites based on TiC–(TiB_2_)–Al and TiC–(TiB_2_)–iron-carbon alloy systems. At the same time, it should be noted that the traditional methods of mechanically grinding and mixing these materials in ball mills have significant disadvantages [4,5,6]. In particular, the mechanical grinding and mixing treatment requires a long time, which causes active oxidation of the powder (even in the presence of a protective medium). In addition, the shortcomings of the method of mechanical grinding should include the low productivity of the process, as well as the contamination of the powder by the material of the milling bodies, which can significantly influence the composition and properties of the sintered materials. One of the effective ways of dispersing the powder is grinding it in a jet mill [2]. In a jet mill, elementary acts of destruction are carried out with the impact of particles accelerated by a stream of gas into each other, or on a fender plate made of a hard material. The contamination of treated powder in a jet mill is considerably lower than in vibration and ball mills, but a large loss of the material that is carried away by the exhaust gas occurs in jet mills; hence, this is their undoubtable disadvantage. The possibilities of jet mills are limited, on the one hand, by the difficulty of accelerating large pieces of material and, on the other, by the need for very high speeds in order to grind fine particles and the complexity of capturing the ground products [4].

Currently, scientific attention is directed to novel methods for dispersing the powders using physical fields. The dispersity of the starting powders and metal carbide is increased by several orders of magnitude compared to traditional mechanical grinding.

The synthesis of Ti–Al–C (TAC) composites involves several stages. The powdered mixture of titanium and aluminum is subjected to high-voltage electrical discharge (HVED) in a liquid hydrocarbon (kerosene) to improve the properties of sintered composites. HVED treatment has been intensively developed in Poland and Ukraine in the last two decades [7,8,9]. It is a unique and complex method with a combination of physical and mechanical impact factors. HVED in liquid is an electrical explosion with rapid energy release (lasting microseconds) in an initially small volume discharge channel paved by a plasma appearing under the influence of a high electric potential between opposing electrodes.

The process of high-voltage electric discharge in the “liquid—processed powder mixture” dispersal system has three stages:The stage of forming a conductive channel that closes the interelectrode gap. This stage begins from the moment of the formation of an electric potential difference between the electrodes of the discharge chamber after the breakdown of the spark gap trigger. At this stage, an extensive network of plasma streamers develops in the working environment.The channel stage, which is characterized by a sharp increase in the discharge current and a rapid discharge of electrical energy in the plasma channel. It begins from the moment of closing the interelectrode gap with a high-conductivity plasma channel. This causes heating of the substance in the discharge channel to temperatures from 2 × 10^4^ to 4 × 10^4^ K and an increase in pressure to values from 3 × 10^2^ to 3 × 10^3^ MPa [8]. The plasma discharge channel expands intensively at a speed of the order of hundreds of meters per second, and compression–tension waves affecting the particles of the processed powder are formed and propagated in the working environment.The post-discharge stage, characteristic of an electric discharge in a liquid. At this stage, a pulsation of the post-discharge steam-gas cavity is observed, which is a source of powerful hydroflows with a speed in the range from 300 to 1700 m/s.

When performing a high-voltage electric discharge in a liquid, the particles of the processed powder are affected by the following factors at different stages of the process [10,11,12]:(a)Electromagnetic field in the near zone of the electric discharge channel and in the discharge gap;(b)Light radiation of the electric discharge channel in a wide spectrum (from infrared to ultraviolet);(c)Complexly shaped pressure pulse in the working fluid, generated during the expansion of the discharge channel;(d)Intense acoustic radiation in a wide frequency band (from infrasound to ultrasound);(e)High-speed hydroflows that occur during the growth and collapse of the post-discharge gas–vapor cavity;(f)Intense three-dimensional microcavitation with a full range of accompanying phenomena;(g)Radicals and radical groups that arise in the liquid during its local decomposition by an electric discharge channel;(h)High temperature of discharge channel plasma.

The impact movement of liquid during the development and collapse of cavitation cavities can destroy non-metallic materials and cause plastic deformation of metal objects placed near the discharge zone. The powerful infra- and ultrasonic oscillations that accompany the HVED additionally disperse already crushed materials, cause resonant destruction of large objects into individual crystal particles, and intensify the chemical processes of synthesis and breaking of sorption and chemical bonds.

The propagation of shock waves in dispersed systems is accompanied by the processes of a sharp acceleration of the flow of chemical reactions, as there is a destruction of the original structure of the components, mutual movement and mixing of components and reaction products, and strong heating in the places of contact of the particles. During shock compression, exothermic reactions with significant energy release can occur in the reactive mixture. The positive effect of the shock wave action is a consequence of the high level of local mechanical stresses and temperatures in the powder particles, which, despite the short duration of the process, causes a sharp intensification of interparticle interactions. This can lead not only to the grinding of powders and an increase in their specific surface, but also to the accumulation of various types of defects, an increase in free energy, which also contributes to an increase in chemical activity.

The use of a hydrocarbon liquid as a working medium during the electric discharge processing of powder mixtures enables the exclusion of their oxidation, which is possible during processing in water. In addition, the conditions for the pyrolysis of hydrocarbons are created during the HVED. As a result of pyrolysis, long hydrocarbon chains are broken with the formation of gaseous hydrocarbons, as well as active hydrogen and solid nanocarbon. Synthesized carbon nanoparticles are able to enter into carbidization reactions with the powder particles. The results of these processes are the dispersion of the initial powder, its mechanical activation, and the synthesis of highly dispersed carbides.

Carbon nanoparticles that appear between the powder particles (in high-temperature zones where microplasma channels occur) during pyrolysis of kerosene in a plasma channel have a clear advantage in the formation of new phases. The research results presented in [13] show that even titanium particles that did not enter the microplasma channels directly and, therefore, did not react with nanocarbon, become covered with titanium carbide particles of ~0.1 μm diameter and smaller after processing.

The fact that the activation of Ti powder with the gradual diffusion of carbon atoms takes place during the electric discharge treatment is evidenced by the change in lattice parameters [13]. For example, with an increase in the specific processing energy of Ti powder from 0.67 to 2.67 MJ/dm^3^, parameter *a* of the hexagonal close-packed titanium lattice steadily decreases from 0.2960 to 0.2918 nm, and parameter *c* steadily increases from 0.465 to 0.481 nm [13]. Thus, the titanium lattice is deformed, stretching in the *c* direction and compressing in *a* direction. Such changes indicate the presence and gradual increase in the concentration of internodal nanocarbon in the crystal lattice of titanium, which gradually leads to the formation of titanium carbide.

When hydrocarbon liquid with sp2-hybridization (namely, kerosene) is used as a working medium for HVED, graphite-type short-range order carbon is synthesized, and using hydrocarbon liquid with sp3-hybridization (dodecane, cyclohexane, or hexane) leads to a synthesis of diamond-like short-range order carbon [10,14,15,16].

The pressure and temperature in the discharge channel of the high-voltage discharge reach 1 GPa and 50,000 K, respectively, during high-voltage discharge. The temperature lowers to 1000 K and the transition layer thickness becomes 2 mm at the interface between the plasma channel and the environment [17]; therefore, during the pyrolysis of kerosenethe, the formation of solid carbon becomes possible [13]. The thermodynamic analysis of reactions that can occur during HVED in the metal powder–liquid hydrocarbon mixture system shows the possibility of the synthesis of processed metal carbides and intermetallic compounds. Experimental studies indicate that HVED in liquid hydrocarbons is accompanied by particle grinding, the synthesis of nanosized carbide components, and the intermetallic phases of the TAC system. This was confirmed by the X-ray diffraction of powder mixtures subjected to HVED in kerosene [18].

Spark plasma sintering (SPS) compacts powder mixtures after HVED; therefore, there is a need to study the complex effects of electrical currents on the dispersion composition of TiC (TiC_2_) and AlC (AlC_3_, aluminum–carbon) compounds. SPS, also called the pressure-assisted pulse energizing process or the pulsed electric current sintering (PECS) process, is a promising technology for innovative processing in the field of new material fabrication in the 21st century. Previously, the SPS method was used mainly to produce ceramic materials and MAX phases [1]. The MAX phase is a triple-system M_N+1_AX_N_ with a hexagonal close-packed structure where M is a transition metal, A is an element of the A subgroup of the periodic table, and X is carbon or nitrogen [19,20]. An essential distinction between these materials is the layered structure of their hexagonal crystal lattices in which the layers of M and A atoms alternate in a certain sequence. This ensures their excellent physical and mechanical characteristics which combine the properties of ceramics and metals. Among the many MAX phases which are currently synthesized, Ti-based MAX phases are of great interest: Ti_2_AlC, Ti_2_AlN, Ti_3_AlC_2_, and especially Ti_2_AlC can be used at high temperatures and corroding conditions owing to their layered structure [19,21,22]. Together with the MAX phases, other reinforcing particles can be obtained—carbides and intermetallic compounds. Therefore, the fabrication of metal matrix composites with TAC in their reinforcing phase structure is a topical scientific task.

SPS (which is also known as field-assisted sintering (FAS) or pulsed electric current sintering (PECS)) is a relatively new method of powder compaction, which is characterized by short holding time (from 5 to 10 min) and few processing steps, which decreases the energy costs of sintering. SPS is based on the simultaneous action of forced mechanical compaction of particles, the spark plasma of electric discharges that occur between the sintered particles, and local heating of the powder particles after the electric breakdown of thin oxide films on their surface. SPS makes it possible to obtain MMCs with increased mechanical and performance properties, as it requires lower temperatures and holding times when compared to conventional sintering methods. The improved mechanical properties of composites depend on setting proper regimes for both HVED and SPS techniques. Understanding the detailed microstructures will allow the determination of regimes and create a new technology for producing TAC composites with the best properties.

The scientific novelty of the following work rests in it presenting the impact of HVED topology and the material and configuration of electrodes on the regularities of synthesis of carbides and intermetallic phases. These are studied in order to determine the main factors that impact synthesis efficiency for their application as anti-friction (lubricant) materials employed in severe conditions, such as temperatures higher than 1200 K, which is crucial for application in jet engines for the aerospace industry. The application of HVED synthesis to new compositions of micro- and nanosized powder mixtures made of mixtures of elemental metal powders, with the subsequent SPS of the obtained powder mixtures, makes it possible to obtain MMCs (including TMcs) with increased strength (by 10–20%), hardness, and wear resistance.

The aim of this study is to investigate the influence of HVED and SPS parameters on the properties of TAC composites using the techniques of scanning electron microscopy (SEM) and energy-dispersive X-ray spectroscopy (EDS).

## 2. Materials and Methods

The samples were prepared at the Institute of Pulse Processes and Technologies of the National Academy of Science, Ukraine (IPPT). A powder containing 85% of titanium and 15% of aluminum was subjected to HVED in liquid kerosene, i.e., 100 g of hard phase with 1.5 L of kerosene (ratio 1:12). The parameters of HVED are as mentioned here: E_1_ = 1 kJ and E_sum_ = 25 MJ kg^−1^. Here, E_1_ is the stored energy, and E_sum_ is the integral (total) energy that is released during the processing of the dispersed system. The electrical circuit of the HVED is presented in Figure 1. The pictures in Fig 1, c represent pressure wave peaks during simulated HVED treatment with a single discharge energy of 1, 0.5 and 0.25 kJ from left to right, respectively. The SPS of obtained powders was performed using an experimental sintering device developed and produced by IPPT (Figure 2). The SPS parameters are listed in Table 1. The temperature of the SPS depends on the current and the temperature generated has an influence on the quantity of synthesized carbides and intermetallic compounds; thus, it is important to determine the optimal parameters of the processes that make it possible to obtain the highest quantity of the mentioned particles, which have not yet been determined.

The sintered samples were mounted on plastic, ground, polished, and etched for the SEM analysis. The samples were polished using colloidal silica suspension OP-S (supplied by Struers) with a mean particle size of 50 nm followed by proper washing. They were observed using a scanning electron microscope (JEOL JSM IT-500HR) equipped with an EDAX Octan Elite EDS analyzer. The EDS spectra were acquired and evaluated using the EDAX TEAM software v4.5. The dimensions of the cylindrical sintered samples were obtained as Ø8 mm × 5 mm.

The XRD analyses were performed on the D8 Advance diffractometer (Bruker AXS, Karlsruhe, Germany) operating at a tube voltage of 40 kV and tube current of 40 mA. The X-ray beam was filtered with Ni 0.02 mm filter to select the CuKα wavelength (λ = 1.5406 Ǻ). Diffraction patterns were recorded in Bragg–Brentano geometry using a fast-counting Bruker LynxEye detector based on silicon strip technology. The specimens were scanned over the range 2θ = 20–90° at a scanning speed of 6°/min using a coupled two theta/theta scan type. The XRD data of the samples were analyzed, and the quantitative determination of the phases was carried out with the Bruker AXS TOPAS v4.1 [24,25] software program using the Rietveld crystal structure refinement method.

The microhardness tests of composite samples were carried out using an Innova test machine. The Vickers hardness test was performed with a load of 2.5 kgf and a dwell time of 10 s. For each sample, 30 indentations were performed, and the hardness values were noted.

## 3. Results

The SEM analysis of the sintered TAC showed that their microstructures consisted of a metal matrix, strengthening particles. X-Ray diffractometry detected TiC and Al_4_C_3_ carbides and an Al_3_Ti intermetallic phase (Table 1), and a few pores (Figure 3a). Some regions of the microstructure have strengthening particles with irregular shapes distributed in clusters (Figure 3b and Figure 4) or higher in magnitude strengthening particles compared with the other samples (Figure 3c). The lower sintering temperature demonstrates the coarse elongated grains of titanium, bigger voids and pores, and the distribution of irregular carbides and intermetallic phases (Figure 3c,d). The SEM photographs were chosen randomly. All microstructures showed that the reinforcing particles are distributed in clusters but not homogenously.

The initial material consisted of 85% powdered titanium and 15% powdered aluminum. The microstructure of the metal matrix shows dark gray and whitish areas (Figure 3a,b). The other experiments reveal the existence of a bimodal microstructure [12] consisting of a mixture of equiaxed (or globular) α phases and a lath structure composed of the plates of α and β phases, as shown in Figure 3a. The gray α phase has two forms: globular, and lath-like. The whitish regions in this image represent the β phase including small secondary α-laths. The globular α phase is formed by the deformation and recrystallisation of the lath α phase present prior to deformation, while the secondary lath α phase is formed by precipitation during cooling from a high temperature [26]. Sample 14 sintered at a high temperature and low current has small globular α titanium grains and wide light areas (Figure 5).

However, the strengthening particles appear to contain voids and cracks (Figure 5). The grain size of the strengthening particles was determined according to the standard ASTM E112 [27] and is equal to G12-G12.5, i.e., 5.6 and 4.7 µm, respectively.

Different areas of the microstructure were examined by EDS. The analysis of spot 1 (Figure 6) marked on a hard strengthening particle shows that the particle is mainly composed of titanium and carbon with a small amount of aluminum by the predominant peaks of titanium and carbon. The calculation of the weight and atomic mass proves that the strengthening particles may be titanium carbides or titanium–aluminum carbides. Similar results were observed in the EDS analysis of the other strengthening particles.

The analysis of Spots 7, 8, and 9 (Figure 6a) shows that globular grains (Figure 3a) consist mainly of titanium and aluminum (Figure 7). Approximately 11% of aluminum can dissolve into α titanium [28]; therefore, the matrix might be composed of titanium grains probably containing dispersed phases such as Ti_2_AlC and Ti_3_AlC [29]. This was proved by testing the hardness of the samples, which was found to be much higher than that of pure titanium (see Section 4). In addition, a small amount of iron was detected perhaps because of the ablation of the steel electrode.

The summary of all EDS analyses is shown in Table 2. Table 2 presents the elemental composition of light and dark gray grains of the matrix and dark carbide particles. Every phase was tested three times.

Sample 14 sintered at a higher temperature formed carbides containing a much higher content of aluminum than that of sample 13 (Figure 8). Figure 8 shows dispersed light precipitates near grain boundaries.

## 4. Discussion

The electric discharge method of powder preparation has great advantages over other methods based on various factors: (a) it achieves a higher degree of dispersion with small contamination by the tool metal; (b) it consumes less time and energy; and (c) it does not threaten the environment. In addition, the HVED method can lead to heterogeneous reactions that change the phase composition, which increases the quality of the material consolidated from HVED-treated powders. In particular, carbidisation can occur when processing metal powders in liquid hydrocarbons because of the interaction of powder particles with nanocarbons of various allotropic modifications that are formed during the destruction of liquid hydrocarbons by plasma.

The presence of reinforcing phases in the microstructures analyzed in this work may be proved by a determination of their Vickers microhardness (HV). The types of these reinforcing phases will be determined in future works. Scientific studies have shown the hardness of α and β titanium to be up to 250 HV, which is approximately equal to 2.5 GPa. (Figure 9) [29]. The samples investigated in this study exhibit higher hardness values (Figure 10). The Vickers hardness of sintered TAC is approximately 490–550 HV, i.e., 4.8–5.4 GPa. This hardness was determined for Ti_2_AlC [20].

Analyzing the hardness of sample 14, a wide range of differences between some points was noticed. A value of 765 HV was obtained as a maximum and it was detected measuring the clusters of carbides and intermetallic phases. In contrast, some minimum hardness values, such as 243 HV and 323 HV, were achieved at some points located away from these clusters. Comparing the samples after subjecting them to SPS, sample 13 had a hardness value of approximately 500 HV as an average. Even though sample 14 has a hardness value higher than 13, there are some points with poor hardness values. Comparing the microhardness of all the samples, it can be noticed that the composition of the initial powder influences the microhardness values. A higher amount of Ti_3_AlC_2_ as well as a temperature above 1000 °C showed the highest microhardness values, i.e., probably a higher number of synthesized carbides and intermetallic phases.

The same test on the TAC composite at 1100 °C for 5 min using the HVED process by Syzonenko et al. [17] reported similar results, but the hardness values were comparatively lower than those obtained in this study. The dark phase has a hardness of 11 GPa, whereas the light phase has a hardness of 13 GPa. Nevertheless, the overall hardness value decreases to 7 GPa because of the nonuniform distribution of particles in the consolidated material.

X-ray diffraction analysis of TAC samples showed that the composites contain TiC, Al_4_C_3_, and Al_3_Ti. The proportions of these phases were calculated by Rietveld analysis. The phases detected are notorious for material strengthening and increasing the heat resistance of composites [30]. Figure 11 presents XRD curves of the two sample series. The predominant carbide was determined to be TiC.

The quality of the refinement using the Rietveld method is quantified by the corresponding figures of merit:Profile residual, Rp, %Weighted profile residual, Rwp, %Bragg residual, RB, %Expected residual, Rexp, %Goodness of fit, χ2 or (GOF)

The parameters of lattices of different phases were calculated and are listed in Table 3.

Analyzing data obtained from the EDX study, it was noticed that the sintering temperature has an influence on the elemental composition of the reinforcing phases (carbides and intermetallic phases) detected by XRD. The Rietveld analysis calculation (Figure 11, amounts of carbides and intermetallic phases) shows that with increasing sintering temperature, the content of carbon and aluminum, existing in carbides and intermetallic phases, increases as well. Conversely, the content of titanium slightly decreases. The dependence of the content of different elements detected in reinforcing particles on sintering temperature is shown in Figure 12.

Quantitative X-ray diffraction analysis of the phases of TAC composites of samples No. G14 and 14, which displayed high hardness values and could possibly contain reinforcing phases, has not yet been performed and is to be the subject of the further investigation of TAC composites, according to the tasks of the joint Lithuanian–Ukrainian project.

The experiments and analysis of TAC materials will be continued. Mixtures of elemental powders of Ti–Al–C system will be used as base objects for the development of new composite materials. Obtaining these materials will make it possible to develop the production of parts for hot jet engine sections, as well as frictional and anti-friction composites hardened with nanoparticle dispersion, based on titanium and aluminum—carbon alloys, which are expected to exhibit increased thermostability (by 10–15%), strength (by 10 to 20%), and wear resistance (by 30–50%) while maintaining ductility. An investigation of reinforcing particles—fullerenes, will be performed for detecting of Cn (*n* = 6, 7) when changing the technological parameters of the HVED and SPS treatment.

## 5. Conclusions

This study investigated the possibility of creating metal matrix composites strengthened by the carbides and intermetallic phases of TAC by SPS consolidation of the charge after the HVED processing of titanium and aluminum powders in kerosene. An experimental plan (method) was developed which allowed us to establish the relationship between the experimental parameters and results. A relationship between the powder processing parameters and the parameters (current, temperature, and holding time) of sintering was obtained.

It was determined that the parameters of processing influence the number of synthesized carbides and intermetallic phases. That was proven by the measurement of Vickers hardness, SEM, and X-Ray analysis. The optimal parameters of HVED and SPS have not yet been determined and this research can be called a pilot study.

The SEM analysis showed the formation of a mixture of α- and β-Ti strengthened with dispersed carbides and intermetallic phases in the shape of globular particles and elongated needles distributed randomly in clusters all over the surface as a matrix. The hard black surface in the shape of small globular and dendritic structures on the matrix are carbides or intermetallic phases which are over-accumulated in some grain boundaries of the samples. The samples have a few pores on their surfaces. The lower temperatures gave more pores and a coarser microstructure. The microhardness test showed that the metal matrix consists of α- and β- phases with comparatively low hardness values up to 250 HV, and the metal matrix is reinforced by intermetallic phases and carbides with hardness values of 500–600 HV.

The grain size of strengthening particles was determined according to standard ASTM E112 and is equal to G12-G12.5, i.e., 5.6 and 4.7 µm, respectively.

XRD analysis showed that TAC samples contain such phases as TiC, Al_4_C_3_, Al_3_Ti, and AlFe (the last one was detected because of the contamination of steel electrode) that strengthen materials and increase the heat resistance of composites. Higher sintering temperatures gave higher TiC content, predominant phase, and less Al_4_C_3_ and Al_3_Ti. By increasing the sintering temperature, the content of carbon and aluminum, existing in carbides and intermetallic phases, increased as well. Conversely, the content of titanium slightly decreased. The lattice characteristics of different phases were determined by Rietveld analysis.

Further investigation of TAC composites will be performed according to the tasks of the joint Lithuanian–Ukrainian project.

## Figures and Tables

**Figure 1 materials-16-05894-f001:**
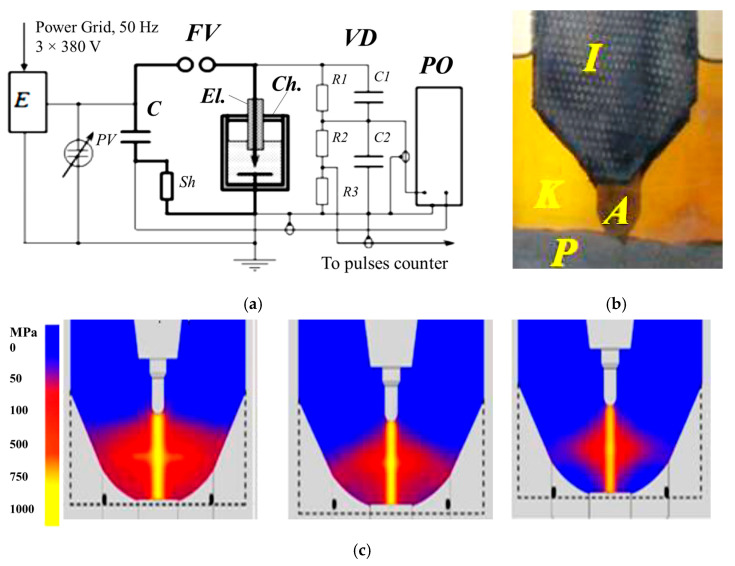
The application of HVED: (**a**) the electrical circuit: E is the power supply; C—energy storage (capacitor); FV—commutator (arrester); VD—voltage divider; PO—storage oscilloscope and control panel, Ch—the chamber with inserted electrode (El.); (**b**) working chambers: I = isolator; A = anode; K = kerosene; P = research powder; and (**c**) pressure wave peaks [23].

**Figure 2 materials-16-05894-f002:**
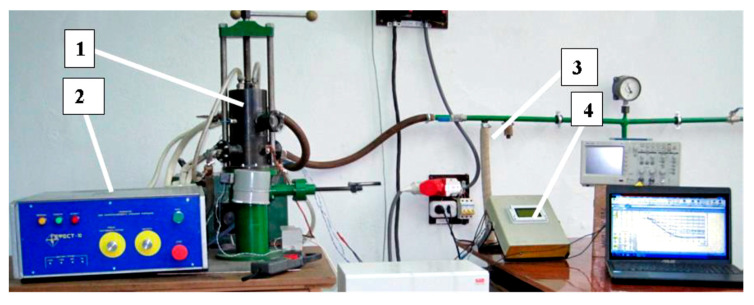
The SPS equipment with the connected electric power source 1 = technological node; 2 = power source; 3 = vacuuming system; and 4 = control system.

**Figure 3 materials-16-05894-f003:**
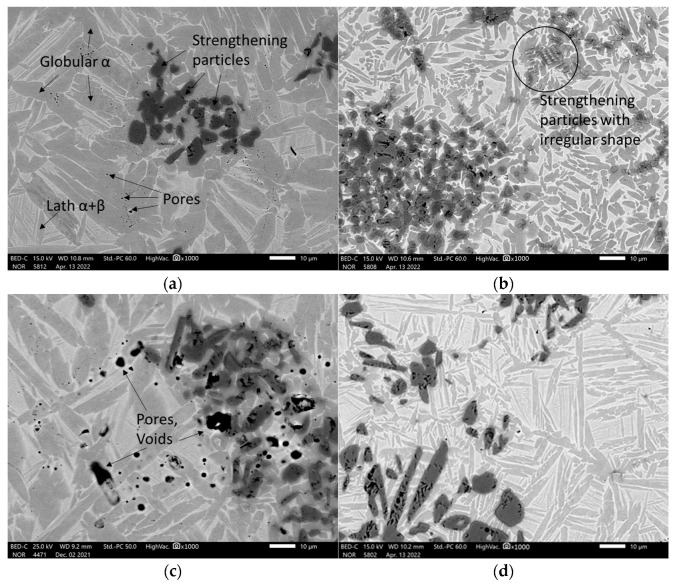
The SEM microstructure of samples: (**a**) the composite with a composition of 75% Ti, 15% Al_3_Ti, 10% Ti_3_AlC_2_; here, the metal matrix and strengthening particles (carbides, intermetallic phases) were sintered at 985 °C; (**b**) the sample with a composition of 70% Ti, 15% Al_3_Ti, 15% Ti_3_AlC_2;_ here, the metal matrix and strengthening particles (carbides, intermetallic phases) were sintered are 1020 °C; (**c**) the sample with a composition of 75% Ti, 15% Al_3_Ti, 10% Ti_3_AlC_2_; here, the metal matrix and strengthening particles (carbides, intermetallic phases) were sintered at 950 °C; (**d**) the sample with a composition of 70% Ti, 15% Al_3_Ti, 15% Ti_3_AlC_2_; here, the metal matrix and strengthening particles (carbides, intermetallic phases) were sintered at 960 °C.

**Figure 4 materials-16-05894-f004:**
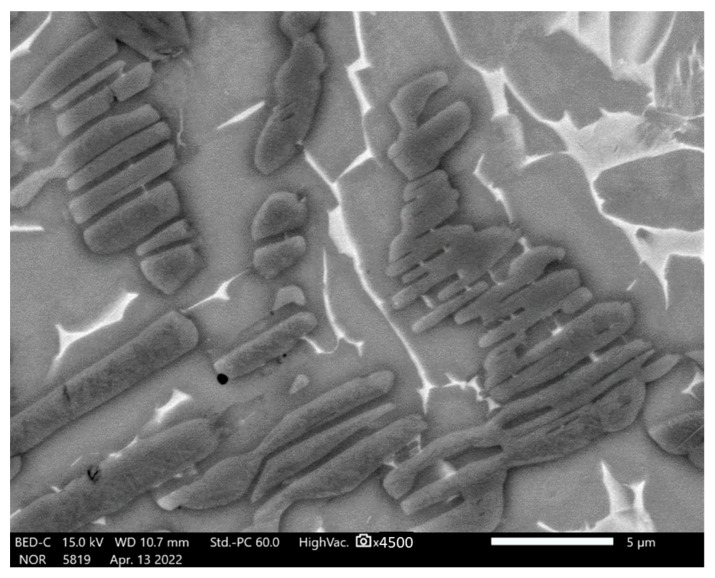
The SEM microstructure of a sample with a composition of 75% Ti, 15% Al_3_Ti, 10% Ti_3_AlC_2_ with particles of irregular shape in the metal matrix.

**Figure 5 materials-16-05894-f005:**
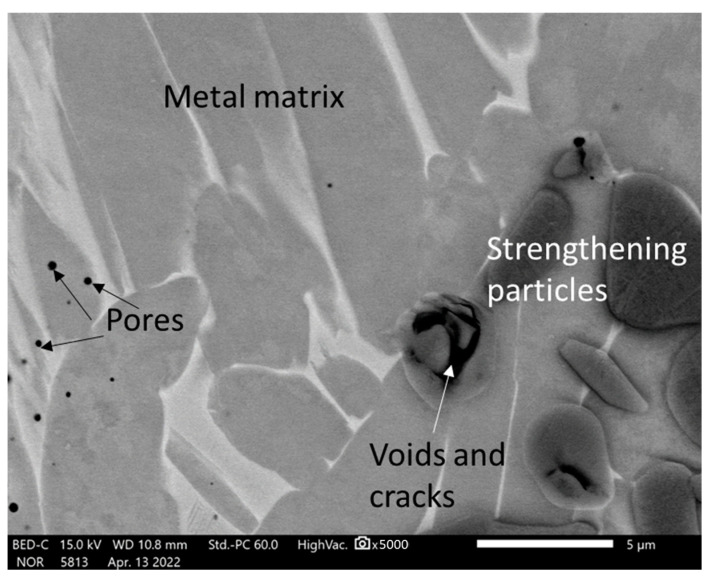
The SEM microstructure of a sample with a composition of 75% Ti, 15% Al_3_Ti, 10% Ti_3_AlC_2_ with the metal matrix and strengthening particles (carbides, intermetallic phases).

**Figure 6 materials-16-05894-f006:**
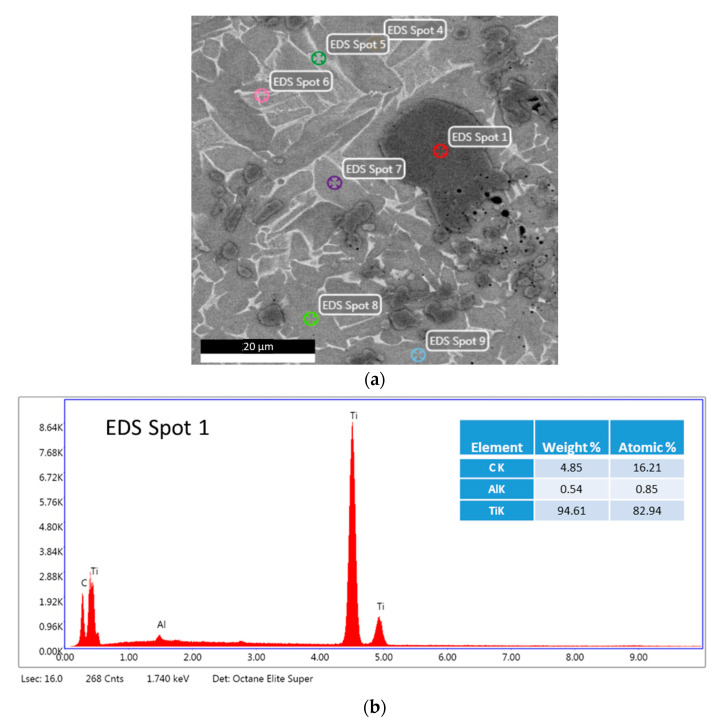
The EDS analysis of Spot 1 of a sample with a composition of 75% Ti, 15% Al_3_Ti, 10% Ti_3_AlC_2_ marked on the dark strengthening particle: (**a**) the microstructure and (**b**) the EDS spectrum of Spot 1.

**Figure 7 materials-16-05894-f007:**
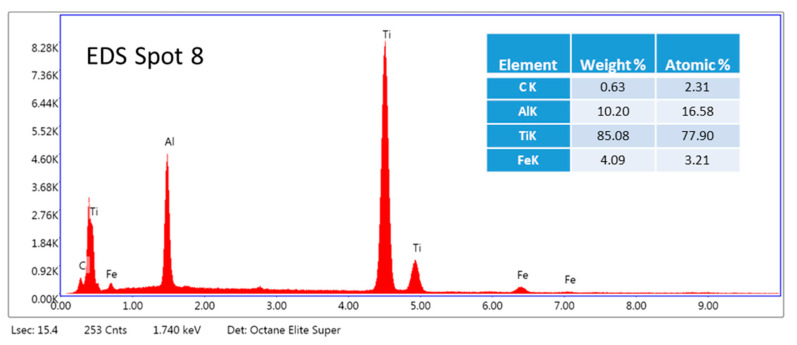
The EDS analysis of Spot 8 of a sample with a composition of 75% Ti, 15% Al_3_Ti, 10% Ti_3_AlC_2_ marked on globular grains.

**Figure 8 materials-16-05894-f008:**
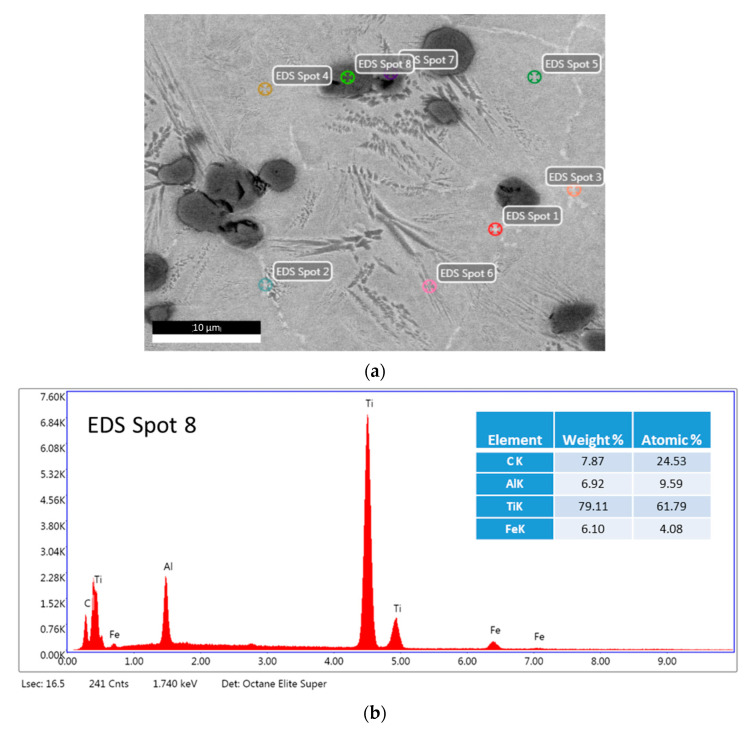
The EDS analysis of Spot 8 of a sample with a composition of 70% Ti, 15% Al_3_Ti, 15% Ti_3_AlC_2_ marked on the dark strengthening particle: (**a**) the microstructure and (**b**) the EDS spectrum of Spot 8.

**Figure 9 materials-16-05894-f009:**
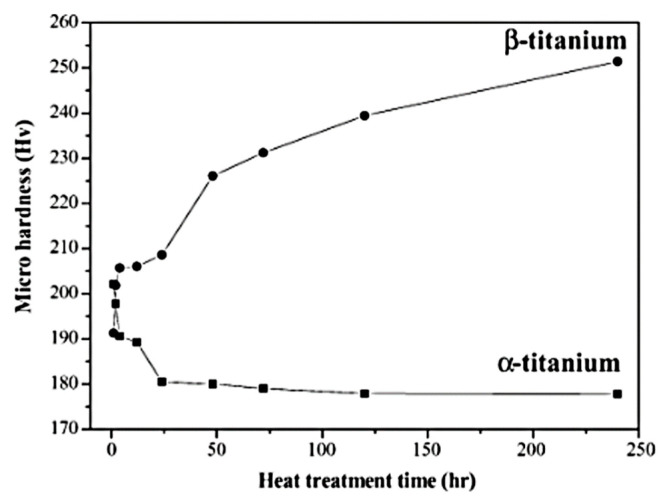
The microhardness variation in α- and β-Ti samples after heating at 750 °C and 1000 °C for different times followed by quenching [20].

**Figure 10 materials-16-05894-f010:**
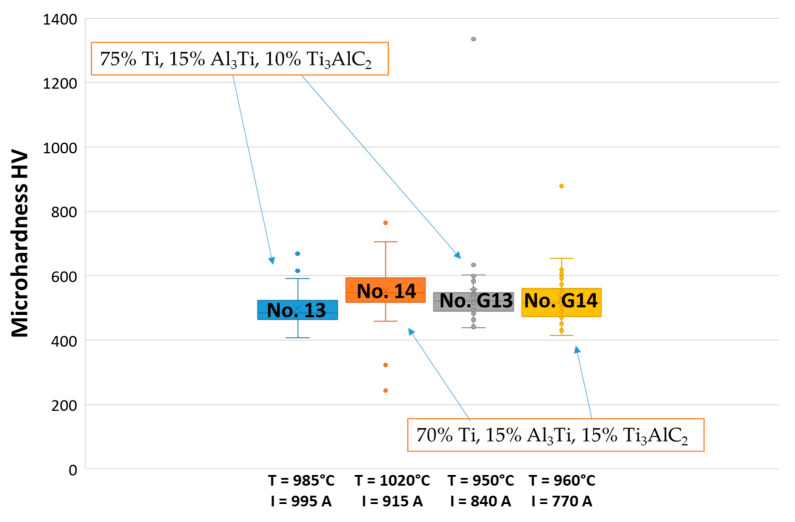
The microhardness of TAC composites with the different initial composition sintered at different temperatures (30 indentations per one sample).

**Figure 11 materials-16-05894-f011:**
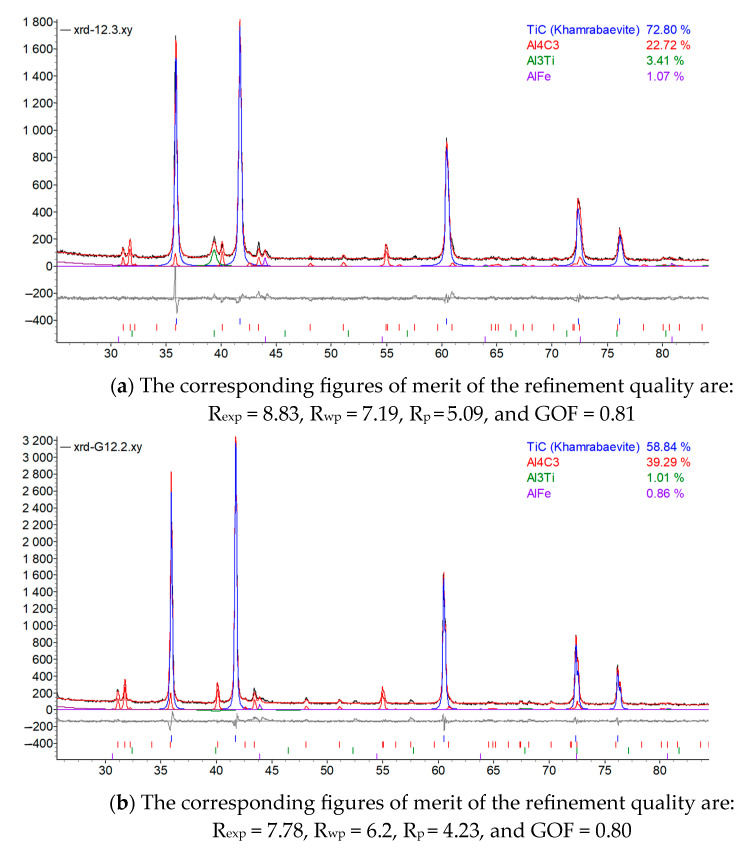
Quantitative analysis of phases of TAC composites with a composition of 80% Ti, 15% Al_3_Ti, 5% Ti_3_AlC_2_ that were sintered by different parameters of the process: (**a**) T = 1005 °C, I = 950 A, t = 5 min; (**b**) T = 995 °C, I = 790 A, t = 5 min.

**Figure 12 materials-16-05894-f012:**
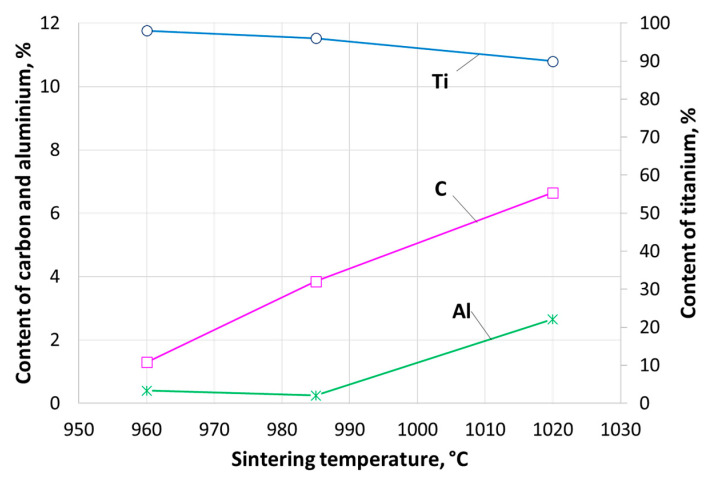
The dependence of sintering temperature on the content of C, Al, and Ti, determined in reinforcing phases of composites.

**Table 1 materials-16-05894-t001:** The parameters of SPS applied during the sintering of samples.

Designation of Samples	Sintering Parameters			Composition of the Powder
	T, °C	I, A	t, min	
G12	995	790	5	80% Ti, 15% Al_3_Ti, 5% Ti_3_AlC_2_
G13	950	840	5	75% Ti, 15% Al_3_Ti, 10% Ti_3_AlC_2_
G14	960	770	5	70% Ti, 15% Al_3_Ti, 15% Ti_3_AlC_2_
12	1005	950	5	80% Ti, 15% Al_3_Ti, 5% Ti_3_AlC_2_
13	985	995	5	75% Ti, 15% Al_3_Ti, 10% Ti_3_AlC_2_
14	1020	915	5	70% Ti, 15% Al_3_Ti, 15% Ti_3_AlC_2_

**Table 2 materials-16-05894-t002:** EDS analysis of samples of light and dark gray grains and dark carbides.

Element	Content of Element in a Phase, %								
	Light Grains			Dark Gray Grains			Dark Carbides and Intermetallic Phases		
	I	II	III	I	II	III	I	II	III
Sintering T = 950 °C, composition 75% Ti, 15% Al_3_Ti, 10% Ti_3_AlC_2_									
C	0.77	0.73	0.81	0.97	0.97	0.97	3.80	3.92	3.99
Al	3.43	3.26	3.25	3.71	3.74	3.49	0.37	0.41	0.22
Ti	92.79	92.57	92.88	94.87	94.72	95.15	95.46	95.31	95.42
Fe	3.01	3.44	3.06	0.45	0.57	0.40	0.37	0.36	0.34
Sintering T = 960 °C, composition 70% Ti, 15% Al_3_Ti, 15% Ti_3_AlC_2_									
C	0.74	0.60	0.54	0.76	0.81	0.75	10.15	12.25	11.73
Al	8.13	8.62	9.08	10.00	9.56	9.99	1.60	0.93	0.83
Ti	85.91	86.16	86.89	88.37	88.92	88.72	86.88	86.26	87.43
Fe	5.22	4.62	3.50	0.87	0.71	0.54	1.37	0.56	-
Sintering T = 985 °C, composition 75% Ti, 15% Al_3_Ti, 10% Ti_3_AlC_2_									
C	0.63	0.84	0.75	0.76	0.76	0.78	4.58	4.06	4.05
Al	10.20	10.54	9.43	11.97	11.70	11.54	0.54	3.41	1.78
Ti	85.08	85.06	82.90	86.04	86.92	87.04	94.61	91.99	93.87
Fe	4.09	3.56	6.93	1.23	0.62	0.64	0.27	0.54	0.30
Sintering T = 1020 °C, composition 70% Ti, 15% Al_3_Ti, 15% Ti_3_AlC_2_									
C	0.73	0.67	0.61	0.60	0.75	0.68	4.25	4.35	4.14
Al	11.29	11.30	10.75	13.27	12.81	13.12	0.72	0.86	1.07
Ti	81.10	81.43	80.65	85.16	83.56	84.98	94.49	94.43	94.25
Fe	6.88	6.60	8.00	0.96	2.88	1.22	0.54	0.36	0.54

**Table 3 materials-16-05894-t003:** Lattice parameters of samples with a composition of 80% Ti, 15% Al_3_Ti, 5% Ti_3_AlC_2_.

Characteristics	Sample 12.3			Sample G12.2		
	Composition 80% Ti, 15% Al_3_Ti, 5% Ti_3_AlC_2_					
	TiC	Al_4_C_3_	Al_3_Ti	TiC	Al_4_C_3_	Al_3_Ti
Lattice parameters, Å	a = b = c = 4.3275	a = 3.3385 c = 24.9919	a = 3.9611	a = b = c = 4.3259	a = 3.3383 c = 24.9932	a = 3.9104
Lattice system	cubic	rhombohedral	tetragonal	Cubic	rhombohedral	tetragonal
Wt%-Rietveld	72.80	22.72	3.41	58.84	39.29	1.01

## Data Availability

Data sharing not applicable.

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
