# Peer review of "Investigation of the Microstructure of Sintered Ti–Al–C Composite Powder Materials under High-Voltage Electrical Discharge"

_materials, 2023, doi:10.3390/ma16175894_

Round 1

Reviewer 1 Report (New Reviewer)

1.      The abstract section is not well organized. The authors should focus the most significant results and conclusion.

2.      The English writing throughout the manuscript need to be carefully checked. For example, In line 32, harden is a verb, here the authors should use a noun hardness.

3.      In line 33, It was said ‘which leads to excellent functional performance’. Generally speaking, use of strengthening particles is to improve the mechanical performance rather than functional performance.

4.      When introducing the features of metal matrix composites and Ti matrix composites in the first paragraph, more latest published papers, for example https://doi.org/10.1016/j.surfcoat.2023.129577 and https://doi.org/10.1016/j.jallcom.2023.169151, are suggested to be cited to better support the paper.

5.      I suggest the authors to shorten the introduction section.

6.      In figure 1, what do VD PO represent. And please indicate electrode in Fig.1a.

7.      The editing and numbering of Fig.3. Fig 3a, 3b, 3c and Fig 3d are separated images. They should be numbered as Fig 3, Fig 4, Fig5 and Fig 6. Otherwise please emerge them into one figure.

Can be improved.

Author Response

Thank you for your valuable remarks trying to make this manuscript better. Please, find the corrections and answers.

Reviewer 2 Report (New Reviewer)

Based on highlighted text and areas, it seems that the authros have addressed the previously provided comments and revision by other reveiwers.

Author Response

Thank you for your valuable time trying to make this manuscript better. Please, find the document with all answers to other reviews.

Reviewer 3 Report (New Reviewer)

The authors made research on investigation of the Microstructure of Sintered TI–Al–C Composite Powder Materials under High Voltage Electrical Discharge ,However all the corrections was made as per reviewer suggestion and it finds satisfactory. Hence it may be considered for acceptance and further publications

Author Response

Thank you for your positive evaluation, for your valuable remarks!

Round 2

Reviewer 1 Report (New Reviewer)

Can be accepted now.

This manuscript is a resubmission of an earlier submission. The following is a list of the peer review reports and author responses from that submission.

Round 1

Reviewer 1 Report

This manuscript focuses on the study of the microstructure of Ti-Al-C composites fabricated with spark plasma sintering. Please find my comments below:

1) Please change TI-AL-C in the title to Ti-Al-C

2) Introduction needs to be expanded. Please include other works on Ti-Al-C composites fabricated with similar processes.

3) Explain the novelty of the work.

4) What is the effect of HVED on the powder microstructure? You expalain the process but not the effect of the process on the material.

5) Please remove Fig3, it is not contributing anything to the results.

6) On SEM figure captions please use the composite composition rather than sample number.

7) You need to identify the strengthening particles. Explain the diffusion processes and expand the microstructure discussion

8) Fig 6 shows poor reinforcement distrubution in the matrix. Is this a random photo? Or there is a consistency of poor homogeneity?

9) SEM images appear to be random. You should put then in order preferebly starting from a low mag to high mag and discuss the microstructural aspects in more detail.

10) Fig 12: Which diffractogram corresponds to which composite?

11) My closing remark is that while the manuscript is interesting, it lacks consistency and cohesion. You are kindly requested to make the appropritate changed that will help the author understand the importance of your work, explan step by step the microstructural evolution and the composite properties.

Minor changes are required.

Author Response

Dear reviewer,

Thank you for your valuable remarks and questions. Please, find the corrections that are marked in yellow. In addition, you will find and other improvements made after all revisions.

Sincerely,

Authors

Reviewer 2 Report

In this study, it is understood that titanium-rich alloys of the Ti-Al-C system are synthesized via spark plasma sintering and high-voltage electrical discharge is used in a composite powder kerosene solution containing 85% Ti and 15% Al to obtain nano-sized particles to be used in this synthesis.

Titanium matrix MAX-phase reinforced samples obtained with different powder compositions and sintering parameters were investigated microstructural and micromechanical.

Although the subject of the study is interesting, there are many uncertain and incomprehensible points in the study.

These are as follows:

1. The study contains scattered information and the data is not presented in a systematic way.

2. Appropriate experimental design was not made to confirm the hypothesis, the relationship between the experimental parameters and the results was not clearly given.

3. The process of processing 85% Ti and 15% Al composite powder with high voltage electrical discharge in kerosene solution to obtain nono size powder and the properties of the powder obtained at the end of this process are not given adequately and clearly.

4. Sample 13 and sample 14 are compared microstructurally, but the production parameters of these samples (temperature, current and composition) are completely different from each other and all parameters change at the same time. It is unclear which parameter causes the structural difference.

5. After sintering, the MAX phase formations (Ti2AlC or Ti3AlC) in the Ti matrix should be determined by X-RAY analysis. Hardness measurement will not be sufficient to evaluate phase formations.

6. In Figure .11, samples 13 and 14 are compared in terms of hardness. Why sample 14 has higher hardness, which process parameter provided it. For these samples, all three sintering parameters change at the same time, the parameter affecting the result is uncertain.

7. Microstructural evaluations were made for samples 13 and 14, but phase analyzes ( Rietveld quantitive phase analysis) were made for samples 12 and G 12. These analyzes should have been performed on samples 13 and 14, and the results should have been evaluated together with EDX analyses.

8. It is not clear on which analysis results Fig 13 was drawn. (For example, there is no analysis data for G 14 sample at 960o C sintering temperature.

Author Response

Dear reviewer,

Thank you for your valuable remarks and questions. Please, find the corrections that are marked in green. In addition, you will find and other improvements made after all revisions.

Sincerely,

Authors

Reviewer 3 Report

This paper deals with Ti-Al alloys reinforced by MAX phases. The topic is interesting and is worth to be investigated. The industrial interest of the compounds presented is rather obvious. 

The reviewer made a bunch of remarks and asked questions that the authors can find inserted as comments in the pdf file attached.

Compilatio software shows 2% of identical sentences with one of the authors work: Syzonenko, O., et al. "High-energy synthesis of metalomatric composites hardened by max phases of Ti-Al-C system." Machines. Technologies. Materials. 12.10 (2018): 395-397 and another 1% of similitude with another work of the authors : (Materials 2022, 15, x. https://doi.org/10.3390/xxxxx www.mdpi.com/journal/materials). This is not huge but the work would be improved if similitudes were avoided. We recommend to rewrite some sentences.

Also, there are 7 over 19 references that are from the authors, >25% of autocitations is too much, even if the authors are top researchers in the field.

The Abstract section must be rewritten since there are several sentences that are incorrect or ambiguous. Moreover, the first sentence is identical (and wrong!) with one of the last paragraph of the Discussion section. 

The material and Methods section must be reinforced with some details: proportion of kerosene, wt. % or vol%, etc. Fig. 1.c show some images that are probably simulations. It must be mentioned. The Fig 1.a. is low quality and must be improved. 

Despite the fact that there are 5 series of samples that are studied here, the Results section presents only few of them and never all the results in XRD, SEM, EDX, microhardness. This is unacceptable. The authors should show all the results or maybe chose to investigate only two series like for example G14 and 14 or G12 and 12. Here, the authors mix different powder compositions with different SPS treatments. Thus, it is very difficult to draw conclusions is several parameters are moving at the same time! If the authors chose to keep all the series, they must give results as tables for XRD phase proportions, EDX compositions and micro-hardness.

In the Discussion section (line 204) there is paragraph that would fit better within the Introduction section, after some minor changes. 

Please find the other comments and remarks in the pdf attached. 

For all these reasons the reviewer recommends Reconsider after Major Revision. 

The author must congratulate the authors for their work in this difficult period of war and hope the best for Ukraine.

The English language should be improved, some of the sentences are ambiguous or incorrect. The best option would probably be to use an online service to improve the quality of the language. 

Author Response

Dear reviewer,

Thank you for your valuable remarks and questions. Please, find the corrections that are marked in purple. In addition, you will find and other improvements made after all revisions.

Sincerely,

Authors

Reviewer 4 Report

Title: Investigation of the Microstructure of Sintered TI–AL–C Composite Powder Materials under High Voltage Electrical Discharge

*    Title:  TI–AL should be corrected.   

*    Abstract: What are the amounts of compounds?    

*   Introduction:  Is the objective of the work only microstructure characterisation?     

*   Results:  Please measure the size of the compounds.             

*     Conclusions:  Can be revised.      

English can be better.

Author Response

Dear reviewer,

Thank you for your valuable remarks and questions. Please, find the corrections that are marked in blue. In addition, you will find and other improvements made after all revisions.

Sincerely,

Authors

Round 2

Reviewer 1 Report

The required changes were implemented. The paper is ready for publication.

I would like to congratulate the authors from Ukraine for keep working during those challenging times. I would like to wish the best for them and their country!

Author Response

Dear reviewer,
We thank you for your work and precious time, but especially for the encouragement to thank all Ukrainians fighting for freedom and independence. In these times of war, it is especially important not to give up and continue the works started. Thank you again.

Sincerely,

Authors

Reviewer 3 Report

The paper was improved but unfortunately all the remarks were not taken into consideration.

There are still many small English mistakes, sometimes the defined article “the” is missing.

Thus, for example in the sentence “The parameters of HVED are as mentioned here: E1 = 1 kJ and Esum = 25 MJ kg−1” it is not clear what E1 and Esum are. Please define them before give values. You can eventually refer to other articles for detailed definitions of those two parameters, but you absolutely must recall here what they designate.

The sentence “The parameters of cylindrical sintered samples were obtained as Ø8 mm x 5 mm” is incorrect since the you speak here about dimensions not about parameters. Replace “parameters” by “dimensions”.

The unfortunate sentence “Rietveld quantitative phase analysis of carbides of TAC samples showed that composites contain TiC, Al4C3, and Al3Ti that strengthen material and increase heat resistance of composites [27]” is still present in the article. First, no Rietveld analysis can show the presence of any phase, the phase detection is independent from the Rietveld analysis and is made prior Rietveld. The Rietveld analysis can calculate the phase proportions once the phases were detected but are certainly not able to show that a particular phase is present in the compound. Second, the fact that these phases strengthen the material has nothing to do with the Rietveld analysis. Please split the sentence in two to say that the Rietveld analysis allow to calculate the phase proportions and in a second sentence to say that these phases are notorious for the material strengthening, eventually.

There is still no values given for the goodness of fit for Rietveld refinement.

In the last paragraph of the Discussion section the percentages announced here for the increase of thermostability, strength or wear resistance are not measured in this article (since in the entire article there is no question of wear, for example), but probably expected percentages. The authors should replace the word "with" with the more suitable expression "which are expected to".

In the Conclusion section, the authors probably wanted to write “coarser microstructure” instead of “courser microstructure”.

The modifications requested here are minor, that’s why I suggests Accept with minor revision.

There are still many small English mistakes, sometimes the defined article “the” is missing.

Author Response

Dear reviewer,

Thank you for the valuable comments. The answers are highlighted in red.

Sincerely,

Authors.
